# Endoscopic Resections for Barrett’s Neoplasia: A Long-Term, Single-Center Follow-Up Study

**DOI:** 10.3390/medicina60071074

**Published:** 2024-06-30

**Authors:** Per Löfdahl, Anders Edebo, Mats Wolving, Svein Olav Bratlie

**Affiliations:** 1Department of Surgery, Sahlgrenska University Hospital, 413 45 Göteborg, Swedensvein.olav.bratlie@vgregion.se (S.O.B.); 2Department of Pathology, Sahlgrenska University Hospital, 413 46 Göteborg, Sweden

**Keywords:** Barrett’s esophagus, ESD, EMR, endoscopic procedures, endoscopy

## Abstract

*Background and Objectives:* Endoscopic mucosal resection (EMR) and endoscopic submucosal dissection (ESD) are both well-established and effective treatments for dysplasia and early cancer in Barrett’s esophagus (BE). This study aims to compare the short- and long-term outcomes associated with these procedures in treating Barrett’s neoplasia. *Materials and Methods:* This single-center retrospective cohort study included 95 patients, either EMR (*n* = 67) or ESD (*n* = 28), treated for Barrett’s neoplasia at Sahlgrenska University Hospital between 2004 and 2019. The primary outcome was the complete (en-bloc) R0 resection rate. Secondary outcomes included the curative resection rate, additional endoscopic resections, adverse events, and overall survival. *Results:* The complete R0 resection rate was 62.5% for ESD compared to 16% for EMR (*p* < 0.001). The curative resection rate for ESD was 54% versus 16% for EMR (*p* < 0.001). During the follow-up, 22 out of 50 patients in the EMR group required additional endoscopic resections (AERs) compared to 3 out of 21 patients in the ESD group (*p* = 0.028). There were few adverse events associated with both EMR and ESD. In both the stratified Kaplan–Meier survival analysis (Log-rank test, Chi-square = 2.190, df = 1, *p* = 0.139) and the multivariate Cox proportional hazards model (hazard ratio of 0.988; 95% CI: 0.459 to 2.127; *p* = 0.975), the treatment group (EMR vs. ESD) did not significantly impact the survival outcomes. *Conclusions:* Both EMR and ESD are effective and safe treatments for BE neoplasia with few adverse events. ESD resulted in higher curative resection rates with fewer AERs, indicating its potential as a primary treatment modality. However, the survival analysis showed no difference between the methods, highlighting their comparable long-term outcomes.

## 1. Introduction

Barrett’s esophagus (BE) is defined as the replacement of the distal esophageal squamous mucosa with the intestinal metaplastic columnar epithelium. It is highly associated with gastroesophageal reflux disease (GERD) and is more common in men, with the frequency increasing with age. BE is associated with a 30-fold increased risk of developing esophageal adenocarcinoma (EA), and the annual risk of cancer development in BE is between 0.12 and 0.78% [1,2,3]. The surveillance and treatment of BE aims to discover, stage, and remove neoplastic lesions at an early stage to prevent further progression into invasive EA.

Esophagectomy has historically been the standard treatment for high-grade dysplasia (HGD) and early adenocarcinoma (EAC) but is highly invasive, with a high mortality, comorbidity, and low QoL [4]. Consensus has now shifted towards less invasive endoscopic resection (ER) techniques.

For Barrett’s lesions containing dysplasia or early cancer, endoscopic mucosal resection (EMR) and endoscopic submucosal dissection (ESD) are both well-established and highly effective treatments. The European Society of Gastrointestinal Endoscopy (ESGE) recommend EMR for lesions ≤ 20 mm with a low probability of submucosal invasion, while ESD is suggested for lesions suspicious for submucosal invasion, those >20 mm, or located in scarred/fibrotic areas [5].

One large single-center study demonstrated EMR’s effectiveness in achieving a 96% complete remission rate for EAC among 1000 patients, with minimal mortality and a 2% rate of major complications [6]. Piecemeal resection is needed in larger lesions, which complicates histopathological staging and increases the risk of local recurrence [7]. ESD provides higher en-bloc and complete R0 resection rates, crucial for accurate histopathological assessment [8,9,10]. ESD is more technically challenging and time consuming and has been associated with more severe complications compared to EMR [11,12]. A randomized study comparing ESD and EMR found no significant difference in the 3-month remission rates, suggesting limited advantages of ESD for lesions treatable using either method [12].

With this study, we aim to compare EMR with ESD during its adoption phase and evaluate the short- and long-term clinical outcomes, which, as to our knowledge, is the first of its kind from the Nordic countries.

## 2. Materials and Methods

### 2.1. Study Design

This single-center retrospective cohort study compares the short- and long-term outcomes of patients treated with EMR versus ESD for BE neoplasia between 2004 and 2019 at Sahlgrenska University Hospital (SUH). The patients were divided into two groups (EMR vs. ESD) based on their initial endoscopic resection.

### 2.2. Study Setting

SUH is the tertiary referral center for advanced upper GI endoscopy and surgery in the western region of Sweden, with a catchment area of 2.1 million people. The hospital has been performing ERs for more than two decades.

### 2.3. Diagnostic Evaluation and Indications

Prior to the treatment decision, the patients underwent high-resolution magnifying endoscopy for the staging and examination of the BE mucosa to determine the lesion characteristics (type, size, and resectability). Targeted biopsies were taken from visible lesions and 4-quadrant biopsies every 1–2 cm from the BE mucosa, according to the Seattle protocol. A CT scan (thorax and abdomen) was performed in cases with histological and/or endoscopic findings suspecting early or invasive adenocarcinoma. These cases were discussed at a multi-disciplinary tumor conference.

Indications for both EMR and ESD during the time period were as follows:BE with visible lesion and/or confirmed EAC, HGD, or low-grade dysplasia (LGD);Diagnostic workup indicated resectable lesion and no sign of metastatic disease.

From 2004 to early 2013, EMR was the only endoscopic treatment for BE neoplasia at SUH. Although ESD was first introduced for gastric lesions in 2008 due to the thicker layer of the gastric wall and the simpler management of potential complications like perforations, its application to esophageal cases began in 2013. By this time, considerable experience and proficiency with the ESD technique had already been developed through its use in gastric procedures. Both methods were used until 2015, when ESD became the prevailing treatment. The choice of method thus depended more on the time period rather than the type and size of the lesion.

### 2.4. Endoscopic Resection Procedure

All endoscopic resections were performed under general anesthesia with tracheal intubation and using carbon dioxide for insufflation. Until 2007, a Fujinon EG-485ZH (Fujifilm, Tokyo, Japan) endoscope without multiple-band imaging (MBI) capability was used, supplemented by methylene blue staining. After this period, high-definition magnifying endoscopy with multiple-band imaging (HDMBI) (Fujifilm, Fujinon EG-590ZW, Tokyo, Japan, with processor and light source series 4400; FICE settings: R = 520 nm, G = 500 nm, and B = 405 nm) was used in the vast majority of the procedures. Approximately 90% of ERs were performed by a single surgeon (A.E.).

EMR was performed using the cap-and-snare technique. The planned resection area was marked with diathermia. Submucosal injection with a viscous solution and adrenaline was applied, the mucosa was sucked into the endocap, and the electrocautery snare was appropriately attached for resection. When required, the procedure was repeated to resect the whole premarked area (piecemeal).

When performing ESD, the planned resection area was marked with diathermia. Submucosal injection with a viscous solution and adrenaline was applied. After the opening of the submucosal layer, the dissection was performed, in the vast majority of the procedures, by using an SB Knife Jr (Sumitomo Bakelite Co Ltd., Shinagawa-ku, Tokyo, Japan, MD-47703W), and the specimen was resected en-bloc.

The patients were observed in the hospital overnight. Proton pump inhibitors (PPIs) were administered orally with a doubled dosage (40 mg × 2). Oral intake was restricted to a liquid diet for 7 days.

### 2.5. Histopathological Assessment

All resected specimens were analyzed at the Department of Pathology at SUH by gastrointestinal pathologists. After resection, the specimens were oriented and pinned on a cork board followed by fixation in 5% buffered formalin. For piecemeal resection, this was carried out during the resection to optimize the histopathological assessment of radicality. After fixation, the specimens were embedded in paraffin, sectioned, and stained with hematoxylin–eosin. Each case of dysplasia was evaluated by two gastrointestinal pathologists. During the initial part of the study period, the tumor differentiation grade and lymphovascular invasion were not systematically evaluated.

### 2.6. Follow-Up

In patients who were referred for surgical treatment, radiotherapy, or entered a palliative stage after post-ER staging, no endoscopic follow-up was performed.

All other patients were included in a surveillance program with a regular endoscopy with histopathological examination. FUs were scheduled after 3, 6, and 12 months. After that period, patients with residual BE mucosa were observed annually. At each FU, the mucosa was examined with HDMBI, and biopsies were taken from the mucosal scar. Mapping biopsies from the residual BE were also routinely taken. Additional endoscopic resections (AERs) were performed using either EMR or ESD to treat synchronous and metachronous lesions as well as local recurrences. The remaining BE mucosa with intestinal metaplasia was routinely treated using radiofrequency ablation (RFA).

### 2.7. Study Data

After the approval from the regional ethics committee (Dnr: 2019-05493), data were collected from medical records for all patients treated with ER of the esophagus between 2004 and 2019. The data were compiled in a study database.

The following data were recorded: patient age and sex, ASA classification, BE classification according to Prague, lesion characteristics according to the Paris classification, procedural characteristics, pathology report, and follow-up data.

### 2.8. Patient Selection

Eligible patients were identified and included from the study database based on the following criteria:Patients (age > 18) who underwent their initial EMR or ESD for either suspected or histologically confirmed BE neoplasia within the time period were included in the study.Patients were excluded if both the pre- and post-histopathologies revealed no presence of dysplasia.

### 2.9. Outcomes

The primary endpoint was the complete R0 resection rate, defined as a combination of R0 resection and en-bloc resection. R0 resection was achieved when the vertical and horizontal margins were free from HGD and/or cancer, regardless of whether the resection was performed piecemeal or en-bloc.

The secondary endpoints were as follows:En-bloc resection rate, defined as the resection of the targeted lesion in one single piece;Curative resection rate, defined as complete R0 resection of HGD, EAC, and Sm1 cancer (≤500 μm) in the absence of poor prognostic factors such as a low differentiation grade and lymphovascular invasion (LVI);Complete remission from neoplasia and recurrences at first endoscopy FU, defined as an R0 or R1 resection of HGD or EAC with no residual or metachronous area of HGD or EAC in biopsies taken from the resection scar and mapping biopsies of the original BE segment;Frequency of AER during follow-up period;Morbidity/adverse events (AEs);30-day mortality;Overall survival, quantified as the percentage of patients remaining alive at 1, 3, 5, and 10 years after the initial ER.

### 2.10. Adverse Events

*Perforation* was defined as an endoscopically visible esophageal wall defect exposing the mediastinal space and/or radiological evidence of extramural gas or fluid collection. Transmural tear was noted but not counted as an AE if the endoscopy showed a deep muscularis propria tear without an esophageal wall hole, clinical mediastinitis, or radiological extramural gas or fluid evidence. Clips were usually applied to transmural tear cases. *Major bleeding events* were defined as those not controllable with endoscopic techniques during the procedure, requiring blood transfusions, or causing delayed bleeding that necessitated further intervention or hospital readmission. *Minor bleedings* that could be managed with endoscopic methods were not considered AEs.

### 2.11. Statistical Analysis

Comparisons between groups for numerical variables were performed using Welch’s t-test and the Mann–Whitney U test, depending on the distribution and variance characteristics of the data. For categorical data, the Chi-square test of independence or Fisher’s exact test was used, with the latter applied when the expected cell frequencies were less than five. Survival analysis was performed using Kaplan–Meier curves and the log-rank test (Mantel–Cox test) to compare the survival rates between groups. A *p*-value of less than 0.05 was considered statistically significant.

The multivariate Cox proportional hazards model was utilized for adjusting potential confounders. In the model, we included the age (≤69 vs. >69 years), ASA classification (ASA 1–2 vs. 3–4), treatment group (EMR vs. ESD), and histopathology (HGD or less vs. T1 or worse) as prognostic factors. These variables were specifically chosen to adjust for key differences and confounders between the treatment groups, with each expected to significantly influence survival outcomes: the age and ASA reflect the patient’s health and operative risk, the treatment type compares the interventions’ effectiveness, and the histopathology distinguishes disease severity.

## 3. Results

### 3.1. Patient Characteristics

A total of 117 patients who underwent ER in the esophagus or the gastroesophageal junction between 2004 and 2019 were identified. Of these, 95 patients (85 men and 10 women) met the inclusion criteria for this study. A total of 67 patients were treated with EMR and 28 patients with ESD. The patient demographics and clinical characteristics are summarized in Table 1. The EMR group was younger on average (64.5 years) compared to ESD (70.6 years), with a significant age difference (*p* = 0.012). The gender distribution and ASA classifications showed no statistical significance between groups. The BE segment lengths were not significantly different (*p* = 0.305 for <3 cm vs. ≥3 cm segments).

### 3.2. Lesion Characteristics

The lesion characteristics are summarized in Table 2. Lesions were larger in the ESD group compared to the EMR group, 16.9 mm and 10.6 mm, respectively (*p* = 0.003). The lesion-type distribution, Paris classification, and pre-ER histopathology did not differ significantly between the groups. Most lesions were located between the 0 and 6 o’clock positions, with no significant difference in the vertical distance from the GEJ.

### 3.3. Procedural Characteristics

The procedure characteristics are summarized in Table 3. Success rates were high for both EMR (97.0%) and ESD (92.9%), with no statistically significant difference (*p* = 0.58). ESD achieved a higher en-bloc resection rate compared to EMR, 88.4% vs. 24.6% (*p* < 0.001). The procedural duration was notably longer for ESD, averaging 165 min compared to 83 min for EMR (*p* < 0.001).

### 3.4. Complete R0 Resection Rate

The histopathological outcomes of resected specimens are summarized in Table 4. The R0 resection of neoplasia was 56.8% and 75% for EMR and ESD, respectively. The complete R0 resection rate for neoplasia was 16% (8/51) in EMR versus 62.5% (15/24) in ESD (*p* < 0.001). The curative resection of neoplasia was significantly higher in the ESD group at 54% (13/24) compared to 16% (8/51) in the EMR group (*p* < 0.001), with notable differences in treating high-grade dysplasia and intramucosal adenocarcinoma.

### 3.5. Follow-Up

The follow-up data are summarized in Table 5. There was a significant difference in the post-ER treatment decisions between groups. Specifically, 12 patients (18%) were directly referred to surgery after EMR, compared to 1 patient (4%) for ESD. Additionally, four patients (14%) were directly referred to radiotherapy after ESD, in contrast to one patient for EMR.

A total of 50 out of 67 patients (75%) in the EMR group and 21 out of 28 patients (75%) in the ESD group had an endoscopic follow-up. The mean follow-up time was longer for EMR compared to ESD (79.8 ± 48.2 months vs. 49.2 ± 27.4 months, *p* = 0.002), as were the median follow-up times (90 months vs. 61 months, *p* = 0.011).

There was no significant difference in the complete remission rates at the first FU between the groups (EMR: 20/35, ESD: 16/19; *p* = 0.069). For EMR, 22 patients required an AER compared to 3 in the ESD group (*p* = 0.028). Additionally, the total number of AER procedures was significantly higher in the EMR group than in the ESD group (36 vs. 4, *p* = 0.026). There was also a significant difference in the rate of ablation between the two groups, with 70% in the EMR group versus 38% in the ESD group (*p* = 0.017).

### 3.6. Additional Endoscopic Resections

Details about the AERs are shown in Table 6. EMR was used as an AER in 23 procedures with a mean age of 62.4 years, while ESD was used in 17 procedures, with these patients being older on average, at 70.6 years. High-grade dysplasia was the most frequent pre-ER histology in both groups. The success rates for resections were 100% across both methods. ESD had a longer mean procedure time, at 164 min compared to 71 min for EMR, and achieved a higher en-bloc resection rate (84% vs. 35%). Complete R0 resection and curative resection rates were significantly higher for ESD (61%) compared to EMR (11%).

### 3.7. Adverse Events and Mortality

Table 7 and Table 8 summarize the adverse events and 30-day mortality related to EMR and ESD treatments. Table 7 describes these outcomes by study group: EMR (*n* = 67) and ESD (*n* = 28). There was one perforation (1.5%) and two strictures (3%) in the EMR group and one perforation (3.5%) and one stricture (3.5%) in the ESD group, with one 30-day mortality recorded only in the EMR group (1.5%). Additionally, there were five minor adverse events involving transmural tears, four during ESD procedures and one during EMR, all successfully managed with clip application.

Table 8 expands the analysis to include both the first and additional ERs, a total of 135 procedures. Under this comprehensive view, EMR had two bleedings, one perforation, three strictures, and one 30-day mortality event. For ESD procedures, there were two perforations and two strictures and no observed 30-day mortality.

Two perforations occurred during ESD procedures. One was managed perioperatively with a Danis stent and clips, leading to the cancellation of the procedure. This patient was later found to have invasive adenocarcinoma and was referred for radiotherapy. The second case involved postoperative fever and elevated CRP, with a CT scan revealing pneumomediastinum. This perforation was treated successfully with antibiotics. Two patients developed post-ESD stricture; one was effectively treated with balloon dilation at a follow-up. The second patient, with multiple metachronous lesions, underwent five AERs (1 EMR and 4 ESD) and is currently under treatment for post-ESD stricture.

For EMR, there was one perforation, treated successfully with a Danis stent. Two instances of postoperative bleeding and three cases of post-EMR stricture were reported. One patient, treated for HGD, was upgraded to adenocarcinoma after developing a stricture at the first follow-up; this patient had additional EMR before referral to radiotherapy. Another patient experienced strictures following EMR and subsequent ESD treatments. Additionally, two patients suffered myocardial infarctions postoperatively, with one patient (ASA 4 with T1a adenocarcinoma) succumbing to complications, resulting in a 30-day mortality rate of 1.5%.

### 3.8. Overall Survival Rates

For EMR, the rates were 96%, 84%, 76%, and 62% at 1, 3, 5, and 10 years, respectively. The ESD rates were 86%, 68%, and 64% at 1, 3, and 5 years, with no 10-year data.

The overall survival, stratified by the treatment method, ASA classification, age, and histopathological findings, is illustrated by the Kaplan–Meier curves (see Figure 1a–d). Figure 2 shows the overall survival among all patients, with a detailed stratification based on the post-ER histopathology.

In the subsequent stratified Kaplan–Meier survival analysis (Table 9), these factors were evaluated for their impact on patient outcomes with the log-rank test (Mantel–Cox test). While the type of treatment group (EMR vs. ESD) did not significantly affect the results, both the ASA classification (ASA 1–2 vs. ASA 3–4) and age (≤69 vs. >69 years) showed significant effects on the survival. Specifically, higher ASA scores correlated with lower survival rates, and similarly, older age groups also demonstrated a negative impact on survival. Histopathology (HGD or less vs. T1 or worse), although not statistically significant, showed a trend that suggested potential effects on survival outcomes.

In the multivariate Cox proportional hazards model evaluating the survival time for EMR and ESD, the same prognostic factors were examined. The treatment method (EMR vs. ESD) was not significantly associated with survival outcomes. The ASA classification was a significant predictor of survival; patients with ASA scores of 3–4 exhibited a higher risk of mortality compared to those with ASA scores of 1–2. Age was also a significant factor, and individuals older than 69 years had an increased risk of mortality. The impact of post-ER histopathology (HGD or less vs. T1a or worse) on survival showed a trend toward significance. Detailed statistical results are presented in Table 9.

## 4. Discussion

The optimal endoscopic resection method for Barrett’s neoplasia, whether ESD or EMR, continues to be a subject of debate among clinicians. The aim of this study was to evaluate and compare the short- and long-term outcomes of EMR and ESD during its adoption phase in treating BE neoplasia. This comparison encompassed not only the technical and histopathological outcomes of each method but also the survival analysis.

In a single-center study evaluating EMR for EAC, the complete remission rate was 96% in 1000 patients [6]. There were no mortalities, and the number of major complications was low (2%). Another study showed similar results, with complete remission in over 95% when using EMR in both HGD and EAC [7]. The main limitation of EMR is the need of piecemeal resection for larger lesions, which aggravates the histopathological assessment. Piecemeal resection has also been associated with a higher risk of local recurrence [7]. Proponents of EMR argue that the risk of recurrence can be reduced by routinely examining all of the remaining Barrett epithelium after ER, as recommended by guidelines [13].

ESD is more technically challenging and time consuming, with a steeper learning curve [11], but has the advantage of high rates of en-bloc and R0 resections that allow for accurate histopathological staging. ESD has been associated with more severe complications compared to EMR [11,12]. Yang et al.’s meta-analysis reviewed 11 studies on ESD for BE neoplasia, involving 501 patients. It reported a 92.9% en-bloc resection rate, with R0 resections at 74.5% and curative resections at 64.9%. Complications were low, with 1.5% perforation, 1.7% bleeding, and 11.6% stricture rates. The recurrence rate was minimal at 0.17% after an average follow-up of nearly 23 months [8].

Another meta-analysis comparing ESD and EMR found that for lesions over 20 mm, ESD had higher en-bloc and curative resection rates and a lower local recurrence than EMR. The complication rates were similar between the two methods [9]. In contrast to ESGE, JGES guidelines favor ESD for BE, highlighting its superior outcomes in a systematic review: en-bloc resection rates of 96.4% and R0 rates of 81.9% with ESD, versus 50% and 39.7% with EMR. ESD also shows a lower recurrence rate (2.5%) compared to EMR’s 12.4%, with comparable complication rates. Thus, JGES recommends ESD for lesions suitable for ER [10]. A randomized study comparing ESD and EMR in patients with HGD and EA found no significant difference in complete remission at the 3-month follow-up despite a higher rate of R0 resection, suggesting limited advantages of ESD for lesions treatable using either method [12].

This study includes patients with BE neoplasia treated with either EMR or ESD as their first treatment during two slightly overlapping time periods with consistent treatment indications. The vast majority (91%) of procedures were conducted by the same endoscopist (A.E.). This facilitates a direct comparison between EMR and ESD in managing BE neoplasia, a comparison scarcely addressed in prior research.

At our hospital, ESD in the esophagus was first implemented in 2013 after establishing the method through ESD procedures on gastric lesions. The literature suggests that an endoscopist typically needs to complete between 30 to 80 procedures to effectively overcome the learning curve and achieve early proficiency [14,15]. Thus, when evaluating ESD outcomes in this study, it is essential to consider the early implementation phase, characterized by potentially higher complication rates, extended procedure durations, and possibly reduced rates of success, R0 resections, curative resections, and technical efficacy.

The primary endpoint of our study, complete R0 resection, was chosen to provide a robust measure of immediate technical success in EMR and ESD treatments. This endpoint not only facilitates accurate histopathological assessment but also, with the higher rates observed in ESD, may explain the reduced recurrence and lower need for additional treatments compared to EMR. Aligning with previous research, such as the randomized trial by Terheggen et al., it enables meaningful comparisons. Furthermore, the choice of this primary endpoint was particularly relevant given our sample size, allowing for a valid statistical analysis.

Another potential primary outcome could have been the incidence of local residual and recurrent neoplasia. However, both ESD and EMR, followed by RFA, demonstrate low recurrence rates. Selecting this outcome would have necessitated a much larger sample size due to these low rates. Additionally, the retrospective nature of data extraction poses significant challenges in distinguishing between synchronous, metachronous, and recurrent lesions.

To address these issues, we incorporated surrogate markers as secondary outcomes to indirectly measure the incidence of local residual disease, metachronous lesions, and recurrent neoplasia. These markers include complete remission from neoplasia at the first follow-up endoscopy and the frequency of additional endoscopic resections. This comprehensive approach ensures that our study not only measures the immediate technical success but also effectively evaluates the long-term efficacy of the treatments. By doing so, we maintain a balanced consideration of both technical success and clinical relevance, reflecting the nuanced complexity of assessing treatment outcomes in real-world settings.

The patient characteristics detailed in Table 1 indicate a significant age difference, with ESD patients averaging 70.6 years and EMR patients 64.5 years. This difference could be influenced by the distinct time periods in which the treatments were applied: EMR from 2004 to 2015 and ESD from 2013 to 2019. It could also reflect a clinical shift towards more frequent endoscopic treatments in older patients due to changing demographics, the growing confidence in the safety and efficacy of these less physically demanding procedures, and, therefore, increased enrollment and continued surveillance of elderly BE patients. The gender distribution, ASA classification, and the extent of BE mucosa according to the Prague classification showed no significant variance among the 95 patients across both groups.

Lesion characteristics between the EMR and ESD groups revealed significant differences in the size, with lesions treated using ESD being larger on average compared to those treated using EMR, as detailed in Table 2. This difference could be due to selection bias, with larger lesions increasingly being treated with ESD over time, reflecting the growing confidence and capability in handling complex cases previously considered unsuitable for EMR. In the EMR group, “irregular mucosa” types were predominantly documented, which was due to the inconsistent documentation of the Paris classification in the study’s early years. HGD and intramucosal adenocarcinoma were the predominant pre-ER histopathologies for both groups.

The procedure characteristics (Table 3) showed high success rates for both EMR and ESD, at 97% and 92.9%, respectively. ESD procedures were significantly longer than EMR (165 vs. 83 min, *p* < 0.001), likely due to the larger lesion sizes and the technical complexity of ESD. The longer durations for both methods compared to other studies could be attributed to the early adoption phase of ESD, the inclusion of thorough endoscopic evaluations during the operating times, and the non-use of water-jet assistance, which has been shown to reduce the procedure time [16].

The histopathological outcomes (Table 4) showed a significant difference in the post-ER histopathology, with ESD having a higher incidence of adenocarcinoma, including intramucosal types, compared to EMR. This suggests that after the introduction of ESD, more patients were considered eligible for ER, possibly due to its perceived capability to handle more advanced lesions effectively. ESD achieved a higher complete R0 resection rate for neoplasia (62.5% vs. 16% for EMR, *p* < 0.001) and a significantly higher curative resection rate, consistent with findings from a randomized trial comparing both methods for HGD and EAC [12].

The follow-up data (Table 5) showed that equal proportions of patients underwent an endoscopic follow-up post-EMR and -ESD, but there was significant difference in the post-ER treatment decisions. More EMR patients were referred to surgery, while ESD patients were often referred to radiotherapy. This likely reflects a shift in treatment paradigms over time in our hospital, especially in managing EAC post-ER. EMR-treated EAC patients frequently underwent surgery, likely due to EMR’s perceived limitations and the histopathological uncertainty associated with piecemeal resections, influencing clinical decisions towards more referrals to additional surgical interventions for a more definitive curative outcome. Podboy et al. support this, noting greater pathological uncertainty with EMR compared to ESD, complicating definitive diagnosis and staging [17]. In contrast, ESD, often viewed as more definitive, led to fewer surgical referrals and more radiotherapy referrals, particularly among older patients with more comorbidities.

The follow-up data also show differences in the long-term management and outcomes. The EMR group had an average follow-up period of 79.8 months, longer than the 49.2 months for ESD. This difference is due to EMR procedures being carried out earlier in the study period, resulting in longer follow-up times for these cases. Interestingly, despite the significant differences in the complete R0 resection and curative resection rates, there was no significant difference in the proportion of patients achieving complete remission of neoplasia at the first follow-up.

During endoscopic follow-up, more patients in the EMR group underwent AERs compared to the ESD group (44% vs. 14%, *p* = 0.028). This aligns with findings from a similar retrospective study by Mejia Perez et al. [18]. The total number of AERs was significantly higher in the EMR group than in the ESD group (36 vs. 4, *p* = 0.026), despite higher ablation rates of 70% in EMR versus 38% in ESD (*p* = 0.017). Although the extended follow-up might suggest a potential underestimation of metachronous lesions and local recurrences in the ESD group, the median days until AER were 246 for the EMR group and 726 for the ESD group. This timeline indicates that these lesions would likely have been detected within the follow-up periods for both groups. This suggests that the higher rate of AER in the EMR group is not only due to longer follow-up times but may also reflect differences in the treatment effectiveness between EMR and ESD. ESD’s technical advantages, such as higher en-bloc and complete R0 resection rates, may provide a more definitive treatment with lower recurrence rates.

Furthermore, RFA was introduced at our hospital in 2009; thus, it was available for most of the study period. Its absence from 2004 to 2008 could have influenced outcomes for the earliest EMR cases, potentially increasing the risk of recurrent and metachronous lesions. During this initial period, 14 patients underwent a total of 19 EMR procedures, and over 50% of these patients received RFA later during the follow-up, after its introduction.

Contrary to our expectations, EMR and ESD used as AERs achieved similar outcomes to initial ERs, with high success rates and technical qualities, and were not more technically challenging or time consuming, despite possible complexities such as mucosal scarring (Table 6).

The safety profiles of EMR and ESD, as detailed in Table 7 and Table 8, show few adverse events and a single case of 30-day mortality in the EMR group involving a high-risk patient (ASA score of 4). This is consistent across both initial and additional endoscopic resections, confirming the safety of both techniques in treating BE neoplasia, aligning with results from other studies [8,10].

The data presented in Table 9 and the Kaplan–Meier curves (Figure 1a–d) describe the survival analysis and overall survival rates following EMR and ESD treatments for BE neoplasia. Despite the early adoption of ESD and the historical tendency to refer T1a patients to surgery in the EMR group, the survival rates between the two groups did not significantly differ. This finding suggests that EMR and ESD may be equally effective in managing BE neoplasia in terms of their long-term outcomes. However, the small sample size of our study limits the robustness of these conclusions.

In the analysis, a Cox proportional hazards model was employed to assess the overall survival rates between patients undergoing EMR and ESD treatments, with an emphasis on evaluating the impacts of several key prognostic factors. This model shows that survival in this cohort is not significantly influenced by the treatment method (EMR or ESD) but is instead significantly related to patient-specific factors such as ASA classification and age. Also, the histopathology variable had a *p*-value of 0.073, indicating a trend that it could affect survival outcome.

In our retrospective study, we implemented strategies to mitigate common limitations, aiming to provide robust insights into the comparison between EMR and ESD treatments for BE neoplasia. Despite these efforts, inherent limitations exist in our study design, and the results should be interpreted with caution. To address selection bias, patients were systematically selected from the study database based on predefined inclusion criteria, enhancing the representativeness of our cohort. However, as described earlier, variations such as differences in the age, lesion size, and the incidence of adenocarcinoma between the EMR and ESD groups suggest some degree of selection and indication bias. Concerning data accuracy, which is often a concern in retrospective analyses, our study likely exceeds the standard for such research. Endoscopic findings were meticulously documented, and in instances of histopathological ambiguity regarding resection margins (Rx resections), cases were conservatively classified as R1 to maintain stringent criteria for radicality.

The study encompasses a considerable timeframe, covering distinct phases of EMR and ESD usage that may influence its findings. Introduced for esophageal lesions in 2013, ESD significantly altered clinical practice. The impacts of evolving technologies and methodologies must be considered when interpreting the results, particularly the shift in 2007 from high-definition magnifying endoscopy without multiple-band imaging (MBI) to high-definition magnifying endoscopy with MBI, which enhanced the image quality and diagnostic accuracy. Notably, only 12 of the 95 patients underwent their procedures using the older, non-MBI technology.

A total of 135 endoscopic resections for Barrett’s esophagus neoplasia were performed during the study period, averaging 8–9 cases per year. Given Sweden’s low population density and the incidence rate of Barrett’s esophagus, such an annual caseload is expected for a Swedish tertiary referral center specializing in advanced upper GI endoscopy during this time period. This low caseload challenges developing and maintaining a high level of expertise, common in the Swedish healthcare system. In our study, an experienced surgeon performed most procedures, supported by a team of gastroenterologists, pathologists, and surgeons. This approach ensures high expertise despite limited volumes.

While the study’s generalizability may be limited by its single-center, single-surgeon design, the consistent documentation, workup, and indications maintained by the same surgeon help to ensure data comparability, low inter-rater variability, and continuity. The study not only shows the long-term results of EMR and ESD but also provides detailed insights into the practical introduction and advantages of ESD over EMR, showcasing the method’s effectiveness and safety even during its early adoption phase.

## 5. Conclusions

In conclusion, this study demonstrates that both EMR and ESD are effective and safe treatments for BE neoplasia, offering high rates of success with minimal adverse events. Despite ESD being in its early adoption phase, it resulted in a higher rate of curative resections with fewer additional procedures, indicating its potential as a primary treatment modality. The lack of significant difference in the overall survival between EMR- and ESD-treated patients underscores the comparable long-term efficacy of both methods. Our findings advocate for the continued use and integration of both EMR and ESD in clinical practice, with the treatment choice tailored to the patient characteristics, lesion specifics, and institutional expertise.

## Figures and Tables

**Figure 1 medicina-60-01074-f001:**
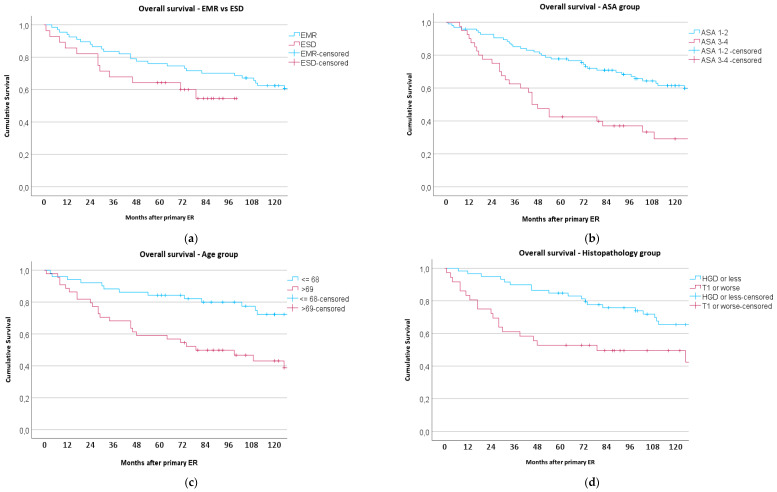
Kaplan–Meier curves showing overall survival, stratified by the treatment method (EMR vs. ESD, *p* = 0.139), ASA classification (*p* < 0.001), age (*p* < 0.001), and post-ER histopathology (*p* = 0.059). The *p*-values in parentheses represent the significance levels from the log-rank test for each category.

**Figure 2 medicina-60-01074-f002:**
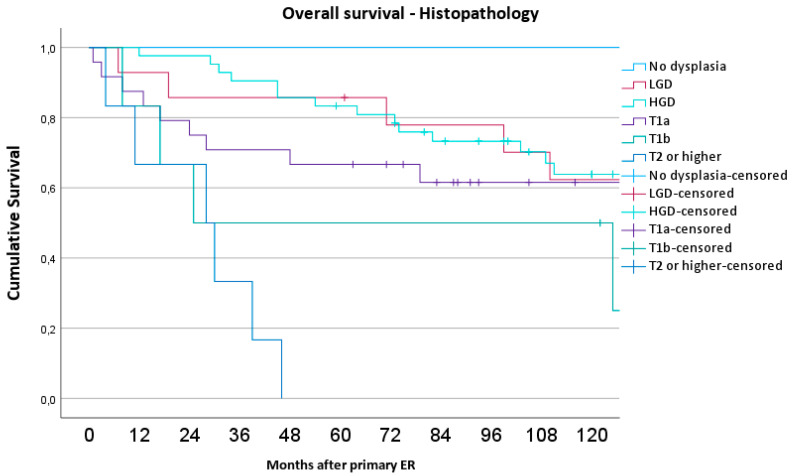
Kaplan–Meier curve illustrating the overall survival among all patients, with a detailed stratification based on post-ER histopathology.

**Table 1 medicina-60-01074-t001:** Patient demographics and clinical characteristics.

	EMR	ESD	*p*-Value
Patients, *n*	67	28	
Mean age, years, mean ± SD	64.5 ± 10.9	70.6 ± 9.3	0.012
Male/female, *n*	59/8	26/2	0.72
ASA classification (%)	*n* = 67	*n* = 28	0.47
ASA score 1	14 (21)	7 (25)	
ASA score 2	34 (51)	10 (36)	
ASA score 3	18 (27)	11 (39)	
ASA score 4	1 (1)	0 (0)	
BE segment, *n* (%)	*n* = 64	*n* = 25	0.305
Short (<3 cm)	17 (27)	10 (40)	
Long (≥3 cm)	47 (73)	15 (60)	
BE length, cm, median (IQR)	*n* = 64	*n* = 25	
Circumferential length	1 (0–6)	1 (0–6)	0.84
Maximal length	4.5 (2–8)	4 (1–7)	0.53

**Table 2 medicina-60-01074-t002:** Lesion characteristics.

	EMR	ESD	*p*-Value
Number of resections, *n*	67	28	
Mean lesion size, mm, mean ± SD	10.6 ± 6.6	19.9 ± 5.2	0.003
Lesion type			0.085
Polypoid	11	8	
Non-polypoid	23	14	
Irregular mucosa	27	5	
Unspecified	6	1	
Paris classification, *n*	*n* = 18	*n* = 23	0.558
Ip	1	1	
Is	3	7	
IIa	7	7	
IIb	3	2	
IIc	2	4	
IIa + IIb	0	2	
IIa + IIc	2	0	
Pre-ER histology, *n* (%)			0.158
LGD	5 (8)	1 (4)	
HGD	51 (76)	17 (60)	
Adenocarcinoma			
Intramucosal	8 (12)	6 (21)	
Submucosal	0 (0)	1 (4)	
Suspected deeper invasion	2 (3)	3 (11)	
No dysplasia	1 (1)	0	
Location, o’clock (%)	*n* = 58	*n* = 27	0.033
0–2	14 (24)	10 (37)	
3–5	30 (52)	10 (37)	
6–8	12 (21)	2 (7)	
9–11	2 (3)	5 (19)	
Vertical location, cm, median (IQR)	1.75 (0–4)	1.5 (0–3)	0.53

**Table 3 medicina-60-01074-t003:** Procedure characteristics and outcome.

	EMR	ESD	*p*-Value
Number of resections, *n*	67	28	
Successful resection, *n* (%)	65 (97.0)	26 (92.9)	0.58
Canceled	2	2	
Duration, mean, min	83	165	<0.001
Resection type, *n* (%)			<0.001
En-bloc	16 (25)	23 (88)	
Piecemeal	49 (75)	3 (12)	
Mean number of pieces *, *n*	3.5		
En-bloc resection rate (%)	24.6	88.5	<0.001
Mean specimen size, mm, mean ± SD			
Length Width	14.9 ± 4.3	35.3 ± 16.9 23.8 ± 10.9	<0.001

* For piecemeal resections.

**Table 4 medicina-60-01074-t004:** Histopathology of resected specimen.

	EMR	ESD	*p*-Value
Number of specimens, *n*	65	26	
Post-ER histopathology, *n* (%)			0.032
No dysplasia/other	2 (3)	0 (0)	1.00
LGD	12 (18)	2 (7)	0.335
HGD	33 (51)	9 (35)	0.244
Adenocarcinoma	18 (28)	15 (58)	
Intramucosal	10	13	0.01
Submucosal (sm1)	0	1	1.00
Submucosal(>sm1)	5	0	0.316
T2 or deeper	3	1	1.00
Differentiation grade, *n* (%)	*n* = 9	*n* = 12	0.401
G1	3 (33)	8 (67)	
G2	4 (45)	3 (25)	
G3	2 (22)	1 (8)	
Lymphovascular invasion, *n*/*n* (%)	N/A	2/12 (16.6)	
R0 resection of neoplasia, *n*/*n* (%)	29/51 (56.8)	18/24 (75)	0.20
Complete R0 resection of neoplasia, *n*/*n* (%)	8/51 (16)	15/24 (62.5)	<0.001
HGD	8/33 (24.2)	7/9 (77.7)	
Adenocarcinoma			
Intramucosal	0/10 (0)	6/12 (50)	
Submucosal (sm1)	0/5 (0)	2/2 (100)	
Invasive	0/3 (0)	0/1 (0)	
Curative resection of neoplasia, *n*/*n* (%)	8/51 (16)	13/24 (54)	<0.001
HGD	8/33	7/9	
Adenocarcinoma	0/18	6/15	
Intramucosal	0/10	6/12	
Submucosal (sm1)	0/5	0/2	
Invasive	0/3	0/1	

**Table 5 medicina-60-01074-t005:** Follow-up data.

	EMR	ESD	*p*-Value
Patients, *n*	67	28	
Post-ER treatment decision, *n*/*n* (%)			0.038
Endoscopic FU	50 (75)	21 (75)	1.00
Surgery	12 (18)	1 (4)	0.099
Brachytherapy	0	1 (4)	0.298
Radiotherapy	2 (3)	4 (14)	0.062
Lost to follow-up/other reason	3 (4)	1 (4)	1.00
Endoscopic FU			
Follow-up time, months, mean ± SD	79.8 ± 48.2	49.2 ± 27.4	0.002
Follow-up time, months, median (range)	90 (2–174)	61 (5–91)	0.011
Complete remission at first FU, *n*/*n* (%)			0.069
HGD	20/33 (61)	7/9 (78)	0.45
Intramucosal	0/2	9/10 (90)	0.045
Total	20/35 (57)	16/19 (84)	
Patients requiring AER, *n* (%)	22 (44)	3 (14)	0.028
Number of AERs, *n*			0.026
EMR	23	0	
ESD	13	4	
Total	36	4	
No. of AERs per patient, *n*			
1	16	2	
2	2	1	
3	1	0	
4	2	0	
5	1	0	
Days until first AER, median, *n* (range)	246 (27–1108)	726 (282–1869)	
Ablation, *n*/*n* (%)	35/50 (70)	8/21 (38)	0.017
Other treatment (during endoscopic FU), *n*/*n* (%)			
Radiotherapy	3/50 (6)	0/21	
Surgery	3/50 (6)	1/21 (5)	

**Table 6 medicina-60-01074-t006:** AER by treatment method (not study group).

	EMR	ESD
Resections, *n*	23	17
Mean age, years	62.4 ± 11.4	70.6 ± 6.7
Pre-ER histology, *n* (%)		
HGD	19	14
Adenocarcinoma		
Intramucosal	3	1
Submucosal	0	0
Invasive	0	2
No biopsy	1	0
Total	23	17
Successful resection, *n* (%)	23 (100)	17 (100)
Duration, mean, min	71	164
En-bloc resection rate (%)	8/23 (35)	14/17 (84)
Histology, *n* (%)		
LGD	4	1
HGD	14	10
Adenocarcinoma		
Intramucosal	4	5
No dysplasia/other	1	1
Complete R0 resection of neoplasia, *n*/*n* (%)	2/18 (11)	9/15 (61)
Curative resection of neoplasia, *n*/*n* (%)	2/18 (11)	9/15 (61)

**Table 7 medicina-60-01074-t007:** Adverse events and 30-day mortality by study group.

	EMR	ESD
Patients, *n*	67	28
Complications, *n* (%)		
Bleeding	0 (0)	0
Perforation	1 (1.5)	1 (3.5)
Stricture	2 (3)	1 (3.5)
30-day mortality, *n* (%)	1 (1.5)	0 (0)

**Table 8 medicina-60-01074-t008:** Adverse events and 30-day mortality by treatment method, including AER.

	EMR	ESD
Resections, *n*	90	45
Complications, *n* (%)		
Bleeding	2 (2)	0
Perforation	1 (1)	2 (4.5)
Stricture	3 (3.5)	2 (4.5)
30-day mortality	1 (1)	0

**Table 9 medicina-60-01074-t009:** Survival analysis (Log-rank test and multivariate Cox proportion hazards model).

	Log-Rank Chi-Square	df	*p*-Value	Cox Model HR (95% CI)	*p*-Value
Treatment method EMR * vs. ESD	2.190	1	0.139	0.988 (0.459–2.127)	0.975
ASA classification ASA 1–2 * vs. ASA 3–4	12.813	1	<0.001	2.281 (1.184–4.393)	0.014
Age ≤69 years * vs. >69 years	16.275	1	<0.001	2.517 (1.256–5.043)	0.009
Post-ER histopathology HGD or less * vs. T1a or worse	3.551	1	0.059	1.833 (0.945–3.556)	0.073

Log-rank Chi-square, df, and *p*-values are derived from the log-rank test used in the stratified Kaplan–Meier survival analysis. Hazard ratios (HRs), 95% confidence intervals (CIs), and *p*-values are derived from the multivariate Cox proportional hazards model. Overall significance of the model: *p* < 0.001 (Omnibus Tests of Model Coefficients). * Reference category.

## Data Availability

The data presented in this study are available on request from the corresponding author due to privacy and ethical restrictions.

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
