# Peer review of "Endoscopic Resections for Barrett’s Neoplasia: A Long-Term, Single-Center Follow-Up Study"

_medicina, 2024, doi:10.3390/medicina60071074_

Round 1

Reviewer 1 Report

Comments and Suggestions for Authors

This is EMR vs ESD study for Barrett's neoplasia in a single center. EMR vs ESD is the common study but the strength of this study is authors analyzed the long-term result, very well written. Below is my comments. 

(Minor Revision)

The author used "compete R0 resection" instead to "curative resection". But in my understanding, many articles use curative resection". Please consider revising. 

Complete R0 resection rate and curative resection rate are very low comparing other studies. Please clarify this reason. 

The author describes a lot of previous study results in the background paragraph. Please simplify background part and move down most of these results to the discussion part. 

Line 144 NCCN guideline recommends ER followed by RFA for background Barrett's, and most of the US centers has been following, however, in this study, only 2 cases in EMR and 4 cases in ESD had RFA. Please explain the reason. 

Comments on the Quality of English Language

Line 84; Please revise "för" to "for".

Line 127. "Sumius JR-knife "Is this commercial name correct? Not "SB Knife Jr Type"? Also, "knife"'s K must be a capital letter.

Table 1. Period (.) must be used rather than comma (,) for most of the table (i.e. "0,012" to "0.012". Please revise it. 

Table 2. Paris Classification 

  Please revise "classification" to "Classification".

  Please revise "IIA, IIB, IIC" to "IIa, IIb, IIc".

Author Response

Best regards

Per Löfdahl

Reviewer 2 Report

Comments and Suggestions for Authors

I congratulate the authors on the effort of collecting and describing all these data from 2004 until 2019. This gives a good insight in the local practice for Barrett’s endotherapy at SUH.

I do have several major comments I would like to address:

-       A retrospective comparison of ESD vs EMR will always be hampered by significant limitations. These studies will suffer from confounding by indication, and different follow-up times. These shortcomings also apply to this manuscript and this is something that needs more attention in the discussion section.

-       The EMR procedures were all performed in the early years of endoscopic therapy for Barrett’s neoplasia, while all ESD’s were performed >2013. This may reflect suboptimal care with less endoscopic expertise and inferior endoscopic image quality in the EMR group. Comparing the two techniques like this is questionable in my opinion. Please address this in the discussion section.

-       During the study period endoscopic eradication therapy for Barrett’s (with the introduction of RFA) greatly improved. This means that during the study period there must have been a major change in management of Barrett’s patients. This could mean that prior to the introduction of RFA patients might have undergone more additional endoscopic resections for metachronous lesions. Please address this issue accordingly.

-       The primary endpoint was the complete R0-resection rate defined as a combination of R0-resection and en-bloc resection. R0 resection was achieved when vertical and horizontal margins were free from HGD and/or cancer. In my opinion this is not the correct primary endpoint. By definition piecemeal resections will be incomplete R0 resections, which probably does not have any clinical implications. The more appropriate primary endpoint would be incidence of local residual and/or local recurrent neoplasia (i.e. HGD or cancer). Please revise this accordingly.

-       The sample size of the study is very limited. Since the currently chosen primary endpoint will by definition yield a big difference (see the previous comment), this difference will be significant despite the small sample size. The study is greatly underpowered to compare the two techniques based on the above mentioned relevant primary outcome. Please address this in the discussion section.

-       Additionally, the authors conduct a multitude of different statistical analyses with this limited sample size. The study is clearly underpowered to address most of these analyses (complication rates, survival) for the EMR vs ESD comparison. Please address this accordingly

-       A total of 117 patients underwent ER for Barrett’s neoplasia between 2004 and 2019. This amounts to less than 8 patients per year. This is a remarkably low case load and could lead to questions about the endoscopic expertise. Please address this in the discussion section.

-       It is remarkable that the mean lesion size for EMR cases was 10,6mm and that the mean number of pieces resected was 3,5. In general lesions up until 15mm will be resectable en-bloc with EMR. It could be that the number of pieces refers to only piecemeal resections, but this is unclear to me. (of note: the asterixes in tables 3 and 5 are not explained in a table legend)

Comments on the Quality of English Language

The quality of the English language is sufficient in my opinion. 

Author Response

Best regards

Per Löfdahl
